# Bond Tests on Clay Bricks and Natural Stone Masonry Externally Bonded with FRP

**DOI:** 10.3390/ma14237439

**Published:** 2021-12-04

**Authors:** Marianovella Leone, Maria Antonietta Aiello

**Affiliations:** Department of Engineering for Innovation, University of Salento, 73017 Lecce, Italy; antonietta.aiello@unisalento.it

**Keywords:** masonry, FRP, bond, Lecce stone, clay brick, mortar joint, effective bond length

## Abstract

Nowadays, the solution of durability problems of existing buildings has a key role in civil engineering, in which there is an ever-increasing need for building restorations. Over the past 50 years, there is a growing interest in a new composite material, fibre-reinforced polymer (FRP), suitable for increasing the resistance and the stability of existing buildings and, consequently, for extending their service life. In this context, the effectiveness of the strengthening system is related to the bond behaviour that is influenced by several parameters such as bond length, the stiffness of the reinforcement, the mechanical properties of the substrate, environmental conditions, etc. This paper aims to analyse the main experimental results from shear tests performed on two kinds of masonry substrates and different types of FRP reinforcements. The purpose is to highlight the role played by many parameters to the bond behaviour of these systems: the mechanical properties of substrates; the stiffness of reinforcements; the type of supports (i.e., unit or masonry unit). The obtained experimental results underlined that the specimens realised with masonry unit show an increase in debonding load and different stress transfer mechanisms along the bonded length with respect to the specimens with a unit substrate. The analysis of the data revealed that the presence of mortar joints cannot be neglected because it influences the interface global performance.

## 1. Introduction

Ancient masonry buildings are very sensitive to seismic events: in other words poor mechanicha tensile proprieties joined to heavy mass. However, a large part of cultural heritage in Italy and many other European countries is constituted by masonry constructions that need repair and retrofitting interventions [1,2,3]. In the past, traditional techniques were used on masonry units such as epoxy injection, crack stitching. chains or wire ropes, masonry or concrete curbs, etc. In this context, the geometrical and mechanical proprieties of the substrate, as well as the unit typology (natural stone, clay bricks, hollow blocks, cement clay interlocking brick, etc.), play important roles in identifying the more effective techniques.

However, in recent decades, the use of fibre-reinforced polymers (FRPs) as externally bonded reinforcement for strengthening masonry constructions has become a well-established technique, aiming to improve the whole structural response of masonry structures or even the mechanical behaviour of some structural elements independently by its typology. More recently the research found new solutions based on inorganic matric, such as FRCM (Fiber Reinforced Cementitious Mortar) or CRM (Composite Reinforced Mortar) [4,5]. FRCM systems consist of bidirectional fibers and inorganic mortar. Their main characteristics are resistance to fire and UV, breathability comparable to that of the support, applicability on wet surface and on rough and irregular support, reversibility of the retrofitting, compatibility with ancient masonry.

However, FRP reinforcements have been proved effective for providing out-of-plane flexural strength [6,7,8,9], in-plane shear and flexural capacity [10,11,12,13,14,15,16], compressive strength of masonry columns [17,18,19,20,21], as well as avoiding local failure mechanisms and strengthening curved masonry elements (vaults, domes, arches) [22,23,24,25]. In many cases, among those mentioned, the bond between the composite and the substrate is particularly relevant, as delamination could constitute the dominant type of failure. The delamination failure is fragile and does not allow the optimal exploitation of FRP performances. For this reason, interface behaviour has been widely studied in order to understand the physical phenomenon and to identify possible solutions able to delay or avoid this kind of failure. Several experimental and theoretical studies are available in the literature dealing with the analysis of the bond between single units and different types of FRPs, but fewer investigations have focused on the bond between masonry units and composite reinforcements. The comparison between the bond behaviour in the case of substrate realized with single units and masonry units, in order to deeply analyse the influence and the role of mortar joints, is reported in [12,26,27,28,29,30,31]. The analysed studies have highlighted that the presence of mortar joints influences the debonding load, the stress transfer mode at FRP–substrate interface, and the effective bond length and failure modes.

Recent studies have shown a direct connection between microstructural and chemical–physical properties of masonry units and mortar and the bond behaviour of the reinforced systems, in terms of porosity, surface roughness, and penetration of resin within the substrates. In fact, in [31], the authors studied the ability of a polymeric matrix to penetrate the substrate, obviously influenced by the properties of the unit surface. In addition, concerning the unit surface roughness, sandblasting pressure treatment has led to the improvement of the penetration and produced an increment of debonding load almost equal to 13% regarding the unit without any superficial treatments.

This paper aims to contribute to the understanding of the influence of mortar joints on the bond between masonry units and FRPs since only a few experimental studies available in the literature discuss these parameters in detail, from both the experimental and analytical points of view. In addition, the available database on the bond is mainly related to specimens with clay masonry unit substrate and only in a few cases to specimens with both natural and concrete masonry units. Within this scope, bond shear tests were carried out on two different substrates, i.e., clay bricks and natural calcareous stone called Lecce stone, in the form of single units and masonry units reinforced with different FRP sheets. The main goal is to investigate the influence of substrates combined with various kinds of reinforcements and the changes caused by the presence of mortar joints. In addition, the effective bond length was determined by an empirical method, and the outcomes were compared with the values obtained adopting the Italian Guidelines [32].

## 2. Experimental Procedure

### 2.1. Specimens and Test Setup

The study reported and described here comprised many experimental tests carried out in order to analyse the bond behaviour between fibre-reinforced polymers and two different kinds of masonry substrates: solid clay units and Lecce stone units. The latter is a calcareous natural stone widespread in the south of Italy. In both cases, shear bond tests were performed on single units and masonry units (i.e., units and mortar joints). Two hydraulic lime-based bedding mortars were used, identified in the following sections as Mortar1, to make clay masonry units, and Mortar2, to make Lecce stone masonry units.

#### 2.1.1. Substrate

The average values of mechanical properties of the two substrates, experimentally evaluated and reported in [10,28], are summarised in Table 1, in which the compressive strength f_cm_, the flexural strength f_t,fm_, and Young’s modulus E_m_ are reported. The corresponding coefficients of variation (CoV) are in round brackets.

FRP Reinforcement: Eight types of unidirectional FRP sheets were used—namely, Carbon1 (CFRP-1), Carbon2 (CFRP-2), Glass (GFRP), Basalt1 (BFRP-1), Basalt2 (BFRP-2), Steel (SRP), Hemp (HFRP), and Flax (FFRP). The tensile strength, f_ft_, elastic modulus, E_f_, ultimate strain, ε_ft_, and the equivalent thickness, t_f_, of the reinforcements were provided by the suppliers and are listed in Table 2. In all cases, epoxy resin was used as the matrix.

#### 2.1.2. Bond Specimens

The specimens made with clay unit substrate, having dimensions of 250 mm × 120 mm × 55 mm, were reinforced by basalt (BFRP-1), carbon (CFRP-1), glass (GFRP), and steel (SFR) composites.

In the case of clay masonry unit substrate, the prisms were built with five units and four mortar joints (Mortar1), about 10 mm thick. Therefore, the final dimensions were 315 mm × 120 mm × 55 mm. The same FRP reinforcements of clay units were also applied to the masonry units.

All shear bond tests were carried out in a single lap configuration with bond length, *L_b_*, equal to 200 mm, and width of the reinforcements, *b_f_*, equal to 50 mm. The width ratio, *b_f_*/*b* is equal to 0.42, where *b* is the width of the substrate. The unbonded length left close to the loaded end of samples was 30 mm. The geometry of bond specimens is represented in Figure 1, by distinguishing between clay unit (Figure 1a) and clay masonry unit (Figure 1b) substrates. The strains along the midline of the bond length were recorded by 7 or 8 strain gauges, glued directly on the FRP reinforcement. The distance between them varied from 4 mm to 40 mm, increasing as they moved further away from the loaded end.

On the stone unit substrate (250 mm × 150 mm × 55 mm), unidirectional sheets of basalt (BFRP-1), carbon (CFRP-1), steel (SRP), flax (FFRP), and hemp (HFRP) were glued. The bonded length was 200 mm. Two different stone masonry units were made: the first type consisted of five units and four mortar joints (Mortar2), with total dimensions of 300 mm × 150 mm × 50 mm, reinforced by BFRP-1; the second type was composed of two units and a mortar bed joint (total dimensions 250 mm × 150 mm × 50 mm) reinforced by both CFRP-2 and BFRP-2. The bond length, *L_b_*, was 200 mm; *b_f_* was 50 mm; the unbonded length at the upper part of the prism was 30 mm.

Therefore, for stone masonry units, the width ratio was *b_f_*/*b* = 0.33. Drawings of the three types of samples are illustrated in Figure 2a–c, respectively, in which the position of the strain gauges is specified. In particular, 8 strain gauges were used for stone units (Figure 2a), while 7 were attached to stone masonry units (Figure 2b,c).

#### 2.1.3. Test Apparatus

The bond tests were performed by means of a universal testing machine using a pull–push scheme [27]. It consisted of a steel frame in which the samples were located and constrained by a reaction system. Steel plates were applied at the end of the reinforcement by means an epoxy in order to improve the grip and reduce the local concentration stress. The test speed was set at 0.3 mm/min until failure.

### 2.2. Summary of Experimental Campaign

All samples were identified by the following labels: the first letter represents the type of substrate (B = clay unit; L = stone unit; MB = clay masonry unit; ML = stone masonry unit); the second letter indicates the type of FRP reinforcement (C1 = CFRP-1; C2 = CFRP-2, G = GFRP; B1 = BFRP-1; B2 = BFRP-2; S = SRP; H = HFRP; F = FFRP); the last number refers to the sequence of the same samples. Only in the case of stone masonry units, the second symbol is preceded by two letters SJ that specify the presence of a single mortar joint.

Part of this experimental campaign has been already reported and discussed in detail in [10,11,28]. The remaining part completes the whole experimental plan, and it is described in detail in the following section. A summary of the experimental campaign is reported in Table 3.

## 3. Result of Bond Tests

In order to better understand the results discussed here in terms of comparative outcomes, a brief overview of the shear bond tests already published in [10,11,28] is reported. Thereafter, with the aim to complete the experimental campaign, other tests were carried out on stone units, and all data are discussed and compared in the following.

### 3.1. Clay Brick Substrate

#### 3.1.1. Clay Unit

The main results obtained from single lap shear bond tests on specimens with clay unit substrate are summarised in Table 4 in terms of axial stiffness of the reinforcements, E_f_·t_f_, average values of debonding load, F_u,deb_, and its corresponding tensile stress, determined by the following equation:(1)$$\sigma_{u,\mathrm{deb}}=\frac{F_{u,\mathrm{deb}}}{b_{f}\cdot t_{f}}.$$

In addition, the failure mode and the efficiency factor, η, obtained by dividing σ_u,deb_ by the tensile strength of the FRP sheet (see Table 2), are reported. The coefficient of variation is in brackets. A part of these data was already reported and discussed in [11].

It is important to underline that the debonding load was not the maximum value recorded during the tests, as already specified in [28], but in some cases, it represented the point of the load–slip curve after which a quasi-horizontal trend could be found. This kind of plateau, in fact, is a clear sign of the beginning of debonding between FRP reinforcement and substrate that may be considered an ultimate condition.

All samples constituted by clay unit substrate exhibited failure mode characterised by a cohesive debonding in the substrate (Table 4) at which it was associated with a relatively low value of *η*. This fact confirmed that the mechanical properties of a substrate play important roles in the debonding mechanism and the failures occurring within the weakest part of the interface. Nevertheless, the values of F_u,deb_ changed according to the kind of fibre reinforcement. As expected, in fact, the higher value of debonding load was reached for samples strengthened with CFRP-1, then SRP, BFRP1, and finally, GFRP, because of the way and the level of load transferred to the substrate, among other geometrical and mechanical parameters, depends on the axial stiffness of the FRP system. This means that, in general, a higher value of E_f_ ·t_f_ corresponds to a higher debonding load.

In terms of load transfer, it is important to compare the strain distribution along the FRP sheet, recorded by strain gauges (Figure 3), which are not discussed in [11]. A specimen for each tested type was considered to draw the curves reported in Figure 3 since a similar trend was found for the other specimens of the same series. At low values of load (Figure 3a,b), the portion of the interface where the shear stress moved from FRP to the substrate, called stress transfer zone (STZ), was predominantly close to the loaded end of the reinforcement for B-B1 and B-G samples. Instead, the transfer zone spread to the end of the bonded length for B-C1 and B-S specimens. Analysing the strain distribution at 0.50–0.75 F_u,deb_, it could be observed that the strain became negligible at about 50–75 mm from the loaded end, in the case of B-G and B-B1 samples, and almost at 150–170 mm from the loaded end for both B-C1 and B-S. In addition, the strain profiles showed that for the specimens reinforced with CFRP and SFRP (i.e., higher stiffness), higher values of strain were recorded, underlining a higher ability to load transfer with respect to the specimens reinforced with BFRP and GFRP sheets.

The average trend of load–slip curves is also reported in Figure 4, confirming the previous considerations—namely, that with increasing the stiffness of the reinforcements, the debonding load increased as well.

#### 3.1.2. Clay Masonry Unit

The experimental data obtained from shear tests on clay masonry units are reported in Table 5, in terms of axial stiffness (E_f_ ·t_f_); bonded length (L_b_); debonding load (F_u,deb_) and corresponding average values (F_u,deb,ave_); tensile stress (σ_u,deb_), according to Equation (1); efficiency factor; slip corresponding to F_u,deb_ (s_u,deb_); average values (s_u,deb,ave_). In the table, the coefficient of variation is reported between brackets. The slip was calculated by integrating the values of strain recorded by the strain gauges applied along the FRP reinforcements, as reported in Figure 1. These data were the results of the last experimental campaign on specimens with clay masonry unit substrate.

The trend of the average debonding load seemed to be linked to the axial stiffness of the reinforcement: the load increased with the increase in stiffness. On the contrary, the efficiency factor, η, reduced with the increase in reinforcement stiffness.

The averaged load–slip curves are illustrated in Figure 5. It can be noted that the average curves, corresponding to prisms reinforced by carbon FRP, exhibited the highest stiffness, compared with other samples. The behaviour was closely linked to the axial stiffness of reinforced systems. In fact, as already reported in Table 5, the highest value of E_f_ ·t_f_ was that of the MB-C1 samples, followed by MB-S, MB-B1, and finally MB-G.

The ultimate condition of the specimens with clay masonry unit substrates was characterised by a cohesive debonding at FRP–substrate interface, with a thinner layer of clay bricks and a thicker portion of mortar joints detached, as shown in Figure 6a,b. A deeper detachment occurred in the first block in the case of MB-S samples (Figure 6c). This last kind of failure was observed with low values of η, ranging from 0.27 to 0.44, and it was also found in other research studies [13,30,32,33,34,35,36,37]. Typically, part of the load applied to the FRP–substrate system was spent for debonding and a residual part for detachment of the wedge of the clay unit. In this case, the high stiffness of SRP led to a better stress transfer between reinforcement and substrate, and therefore, a significant detachment of the clay masonry unit occurred, as shown in Figure 6c. Nevertheless, distinguishing these two contributions was not so completely clear. In addition, it is worth emphasising that, in order to reduce this local effect, an unbonded area was left (30 mm) in the upper part of the substrate [38]. Furthermore, the results could be considered consistent since the scattering among similar specimens was low in terms of both failure mode and debonding load. Moreover, it should be noted that the observed boundary phenomena occurred in both kinds of specimens—namely, the clay unit and the clay masonry unit.

The highest η value was obtained for the three specimens strengthened by glass fibres because, in these cases, the tensile failure of FRP reinforcement occurred (Figure 6d). Generally, the η value is lower in the case of reinforcements with higher performances, as the poor mechanical properties of the substrate cause the cohesive debonding failure long before the exploitation of the FRPs strength. On the other hand, comparing the η values related to the clay unit and clay masonry unit, an increase may be noted. Consequently, as the FRP applied to both substrates was the same, this indicated that the presence of mortar joints allowed an improvement of the interface strength. This improvement was higher for samples with BFRP-1- and GFRP-strengthened systems, i.e., the two most deformable reinforcements.

The strain profiles along the FRP strips are drawn in Figure 7, for specimens with clay masonry unit substrates. In the same figure, the position of mortar layers is also reported, to appreciate their role during the stress transfer process. The evolution of the load transfer process could easily be determined from these strain distribution profiles.

As expected, for the low percentage of debonding load, i.e., (0.25 ÷ 0.50) *F_u,deb_*, the strain–position curves had an almost exponential trend. The influence of mortar joints appeared more evident with increasing the load; in fact, from 0.75 *F_u,deb_*, a pronounced drop in strain occurred, mostly corresponding to the first mortar joint (that close to the loaded end), followed by a rapid rise. The presence of mortar joints revealed a strong discontinuity, leading to a sort of “interlocking” phenomenon that withstood against the debonding between FRP and substrate. On the basis of the results obtained, it seemed that the higher porosity corresponding to the mortar joint promoted a deeper penetration of the resin, as evinced by the thicker portion of material detached at that position, involving an increased stiffness of this part of the substrate with a consequent decrease in the reinforcement strain [38,39,40].

It is worth noting that the values of strain reached for MB-C1 and MB-S were four times less than that recorded for the other two kinds of samples, which were clearly more deformable.

#### 3.1.3. Comparison between Clay Unit and Clay Masonry Unit

Making a comparison between the two kinds of substrates, clay unit and masonry clay unit, an increase in the debonding load from the first to the second type of substrates was generally obtained even with the significant difference depending on the kind of reinforcement. In fact, the highest increment was equal to 99% for samples strengthened by basalt FRP (Figure 8a). On the other hand, the lowest one was 8% for MB-C1 specimens (Figure 8b). In the middle were the specimens reinforced by glass and steel (Figure 8c,d), for which a 50% and 30% increase in debonding load, respectively, was observed. In addition, it is worth noting that generally, the behaviour of the clay masonry unit was more deformable than that of the clay unit, regardless of the type of the FRP reinforcement. This revealed that the mortar joints influence the stress transfer between FRP and clay unit, as previously mentioned.

With the aim to analyse in detail the strain distribution, in Figure 9, the registered strain values for samples with clay unit substrate and specimens with clay masonry unit substrate are represented by histograms. The values, recorded by the strain gauges, were considered at different position along the bond length (at x = 0 mm; x = 50 ÷ 70 mm; x = 180 mm) and at different load levels. This comparison showed that the scatter between strain values of MB and B was more marked in the case of specimens strengthened with low-stiff FRP—namely, BFRP-1 and GFRP. In fact, referring to the strain recorded at x = 0, the specimens with clay masonry unit substrate and reinforced with BFRP and GFRP sheet showed a value of strain higher than 50–55% with respect to that recorded for the sample with clay unit substrate. Otherwise, the strains recorded at x = 0 on specimens with clay masonry unit substrate and CFRP-1-SPR reinforcing materials were 1–3% higher than the corresponding value on the sample with clay unit substrate.

Therefore, with increasing the stiffness of FRP, less influence of the presence of mortar joints was found in terms of strains.

In terms of failure mode, the obtained data showed that for samples reinforced by GFRP, a completely different failure mode was registered from the clay units to the clay masonry units. In fact, the failure of the clay unit was due to cohesive debonding, while for MB-G specimens, the tensile rupture of FRP occurs. As the effective bond length in the case of B-G samples was longer than the unit thickness (>50 mm), higher exploitation of the reinforcement occurred because of the presence of the mortar joint, which allowed further stress transfer, as previously mentioned; finally, failure occurred within the reinforcement once its ultimate strain was almost attained. Referring to specimens with basalt and carbon sheets, the increase in the debonding load was inversely dependent on their axial stiffness (for basalt, it was 13,300 N/mm, and for carbon, it was 40,800 N/mm)—namely, the lowest value of E_f_ ·t*_f_* corresponded to a greater influence of the substrate. Thus, if the reinforcing system was very stiff, the ultimate load appeared unaffected by the presence of mortar joints and their interlocking role, accordingly, as observed before for MB-G specimens. A particular behaviour was observed for steel reinforcement; in fact, it was the stiffest system, but the increment of debonding load was high if compared with that obtained for MB-C1. However, in this case, a slightly different failure mode occurred for the clay unit and clay masonry unit. Indeed, the width of the substrate detached at the end of the shear test was greater for MB-S than for B-S (Figure 10). Therefore, in the presence of very stiff FRP reinforcement, the crack interface into the substrate went deep, and the bed joints of mortar contributed to delaying the debonding. In addition, a partial shear failure of the substrate occurred, as is also evident from Figure 6c.

### 3.2. Lecce Stone Substrate

#### 3.2.1. Stone Unit

The outcomes of shear bond tests on specimens with both stone unit and stone masonry unit substrate discussed in [29] are briefly reported in Table 6, in which the coefficient of variation, when it was statistically significant, is reported in brackets.

Meanwhile, the experimental results of the bond test, performed on the stone unit strengthened with steel, flax, and hemp FRP, are detailed in Table 7. These latter data were the results of the last experimental campaign on specimens with the stone unit substrate.

For the stone unit with steel fibres, a typical cohesive debonding occurred (Figure 11a), while in the case of natural fibres, i.e., flax and hemp, tensile failure of FRP occurred (Figure 11b,c).

Analysing the whole experimental campaign on the stone unit, in terms of failure mode, samples could be divided into two groups: the first group, constituted by L-S, L-B1, and L-C1 specimens, reached the ultimate condition because of cohesive debonding, while for the second one, in which the substrate was reinforced by natural fibres (L-H and L-F), tensile rupture of FRP occurred. In addition, the values of debonding load for the first group were quite similar and almost double those obtained for the second group (Table 6 and Table 7). Finally, still referring to the first group, the very low-efficiency factor corresponded to the ultimate conditions, between 0.12 and 0.31. Slightly higher values were obtained for the clay unit, with the same FRP type and failure mode. This result confirmed that the mechanical properties of the substrate had a significant influence on the FRP–masonry bond. On the other hand, for L-H and L-F, very low values of debonding load were recorded because of the poor tensile strength of both natural fibres, hemp, and flax.

Samples could be distinguished into three categories in terms of load–slip behaviour and longitudinal strains distribution on FRP sheets (Figure 12 and Figure 13, respectively): (1) samples strengthened by high deformable composites, having low values of F_u,deb_ with high slip, and an almost linear trend of the load–slip curve. High strain values could be observed on the FRP zone very close to the loaded end, which became negligible at a distance equal to 20–30 mm (L-F and L-H), showing the low-stress transfer capacity; (2) specimens strengthened by stiff composites (L-C1 and L-S), showing strains level nearly an order of magnitude lower than those recorded for the first group; (3) specimens that exhibited an in-between behaviour (L-B1), compared with other samples, which was characterised by an initial quite stiff behaviour of F–s curve, until around 3 kN, after which the slope of the curve decreased.

This trend fully reflected the different axial stiffness of the FRP reinforcement systems.

Accordingly, in both cases, stone unit and masonry stone unit, debonding of FRP reinforcements was observed, with a thin layer of substrate detached, aside from the tensile rupture of flax and hemp FRP. The boundary effect may be noted when a detachment of a stone wedge at the loaded end occurred, which was often for stone masonry unit samples. Additionally, for L-B1, shear crack occurred, followed by the wedge detachment. Probably, a longer unbonded length should be left at the loaded end. As mentioned before for specimens with clay unit substrate, a part of the fracture energy led to the wedge detachment and the remaining to debonding of the FRP from the substrate.. It may, therefore, be concluded that the local effects did not have a significant impact, considering that the obtained results in terms of debonding loads did not change in any case. F_u,deb_ of samples with stone masonry unit substrate (i.e., ML-B1, ML-B2-SJ, and ML-C2-SJ) was always higher than that of stone unit specimens (L-B1 and L-C1). This confirmed what was already underlined for the bond test on clay unit—namely, the beneficial effect of interlocking of mortar joints.

#### 3.2.2. Comparison between Stone Units and Masonry Stone Units

In Figure 14, the comparison between strain, registered by the strain gauges at x = 0, 60, and 180 mm with increased load, of specimens with stone unit and stone masonry substrate are reported. As can be noted from Figure 14a, the ML-B1 sample showed values higher than ML-B2-SJ and L-B1, and the difference was greater with increased load, especially at 60 mm, from the loaded end of the FRP. This behaviour was closely linked with the presence of mortar joints that caused an increase in deformability of the whole FRP–masonry system. The same results were obtained for samples strengthened with carbon FRP (Figure 14b). Referring to the strain recorded at x = 0 mm, the point discussed previously for clay unit and clay masonry unit substrates was also evident in this case: with the increase in the stiffness of FRP, less influence of the presence of mortar joints was found in terms of strains. In fact, the specimens realised with the stone masonry unit substrate and BFRP showed a value of strain higher than 30–35% with respect to that recorded for the sample with the stone unit substrate. A scatter of 50% was instead observed, comparing the strain values of the specimens reinforced with the CFRP-2 sheet.

## 4. Effective Bond Length

In the last part of this study, a comparison between the theoretical formula proposed in Italian Guidelines [32] to evaluate the effective bond length and what was obtained from the experimental tests is presented and discussed.

The value of the effective bond lengths, directly determined from the experimental results here reported, derived from the following simple assumption: Considering the exponential strain profiles, the distance from the loaded end of FRP reinforcements to that point where the strains are equal to zero corresponds to the effective transfer length. Obviously, that value depends on the level of load applied to the FRP–substrate system. In order to determine the initial effective bond length, it is essential to consider low values of load before the debonding starts. In fact, after that, the transfer length increases with crack propagation [41]. However, it should be noted that the condition of zero strain is not a realistic situation, because the strain–position curves approach zero but often do not reach it. Therefore, it is necessary to define a strain level below which the strain distribution could be considered negligible. For the results here presented, this limit was equal to 2% of ε_max_ for each sample.

In Table 8, the average values of effective bond length, l_ek,exp_, and the corresponding coefficient of variation, are reported for all samples constituted of the units, i.e., clay unit and stone unit. For each specimen, l_ek,exp_ was obtained considering the strain–position curves at 10%, 30%, and 50% of F_u,deb_.

The greatest value of effective bond length and the lowest values of CoV were obtained for SRP reinforcement, for both substrates. On the other hand, samples strengthened with BFRP-1 showed the lowest value of l_ek,exp_. There was, therefore, an evident connection between the effective bond length and the axial stiffness of FRP. Moreover, varying the substrate, l_ek,exp_ was reduced from clay unit to stone unit. This suggested also that a more mechanical performant substrate allowed a better distribution of strain at the FRP–substrate interface, along the bond length. Unfortunately, this method was not suitable for masonry units because of the trend of the strain–position curve at the mortar layer, as analysed previously. On other hand, the strain profiles were influenced by the nonuniformity of the substrate (i.e., unit and mortar) covering the effective transfer length.

The experimental results were compared with those obtained using the relationships reported in the Italian Guidelines [32], considering the characteristic values and then adopting unitary corrective factors (i.e., γ_Rd_, FC) as follows:(2)$$l_{\mathrm{ek}}=\max\left\{l_{\mathrm{ek},1}=\frac{1}{\gamma_{\mathrm{Rd}}\cdot f_{\mathrm{bd}}}\sqrt{\frac{\pi^{2}\cdot E_{f}\cdot t_{f}\cdot \Gamma_{\mathrm{Fd}}}{2}};150\;\mathrm{mm}\right\},$$
where E_f_ is the elastic modulus of FRP reinforcement, and t_f_ is the FRP thickness (Table 2); Γ_Fd_ is specific fracture energy, determinable as
(3)$$\Gamma_{\mathrm{Fd}}=\frac{k_{b}{\;k}_{G}}{\mathrm{FC}}\;\sqrt{f_{\mathrm{cm}}{\;f}_{\mathrm{tm}}},$$
where FC is the confidence factor (assumed unitary); f_cm_ and f_tm_ (= 0.1 f_cm_) are the compressive and tensile strength of masonry substrate, respectively (Table 1); k_G_ is a corrective parameter (k_G_ = 0.031mm in the case of the clay unit and k_G_ = 0.012 mm in the case of the stone unit); $k_{b}$ is a geometrical factor ($\sqrt{\left(3-b_{f}÷b\right)/\left(1+b_{f}÷b\right)}$); f_bd_ is the design bond strength between FRP and masonry, and it is equal to
(4)$$f_{\mathrm{bd}}=\frac{2\Gamma_{\mathrm{Fd}}}{s_{u}},$$
where s_u_ = 0.4 and 0.3 for clay and Lecce stone, respectively.

In [28], the authors have proposed a recalibration of k_G_ factor, based on a larger database of shear bond tests, and they have found that k_G_ is equal to 0.017 mm for the clay unit and 0.011 mm for the stone unit. The values obtained using the new k_G_ are identified as l^*^_ek,1_, while l^*^_ek_ is the maximum values between l^*^_ek,1_ and 150 mm, following Equation (2).

All data are summarised in Table 8.

According to Equation (1), the effective bond length varied from 150 mm to 186 mm for samples with the clay unit substrate and from 150 mm to 205 mm for specimens with the stone unit substrate. It can thus be concluded that, in general, the experimental method proposed, based on strain–position behaviour, underestimated the effective bond length, if compared with the CNR formula (Equation (1)).

Moreover, considering the definition of l_ed_ and the bond length used in this experimental campaign, which was always equal to 200 mm, it can be assumed that the debonding load obtained for both types of samples made with units were the highest load that may be achieved for these kinds of substrates and the used FRP strengthening systems.

A different situation was found using the recalibrated factor [28], which involved an increment of the effective bond length that was more relevant in the case of B-C1 and B-S. Further experimental investigations are recommended by increasing the bond length, in order to further validate the more recently proposed coefficient in the case of stiffer reinforcement.

## 5. Conclusions

In this paper, the bond behaviour of different substrates and various FRP reinforcements was analysed by carrying out single shear bond tests. In particular, two types of substrates were adopted: clay unit and Lecce stone unit, both widely used as building materials in Italy. The tests were conducted on both unit and masonry unit in order to investigate the influence of the presence of mortar joints on the bond performance.

The obtained results allowed the following considerations to be drawn:The stiffness of the reinforcement influenced the stress transfer mechanism at the interface—namely, with the increase in the stiffness of the reinforcement, the debonding load and transfer length increased as well. Based on the obtained results, the stress transfer zone was 1–2 times longer for specimens reinforced with carbon- and steel fibre-reinforced polymer with respect to specimens reinforced with glass- and basalt fibre-reinforced polymer.The stress transfer mechanism between FRP and substrate varied when comparing masonry units and unit samples. In fact, according to the present experimental results, the presence of mortar joints led to a sort of interlocking mechanism between reinforcement and substrate underlined by the presence of strain decays at the same position of the joints. Probably, the different physical and chemical nature of the mortar with respect to the unit promoted a deeper penetration of the resin with a consequent effect of increasing the stiffness of this portion of the substrate.The failure mode occurred in the mechanically weakest component of the FRP–substrate system. Almost all specimens showed a cohesive failure with a thin layer of substrate detached, confirming that the low shear strength of the substrate with respect to that of the reinforcement governed the mode of failure. Only for the specimens reinforced with flax or hemp FRP was a tensile rupture of the reinforcement out of the bonded length observed. In this case, the poor mechanical properties of the natural fibre, especially in terms of stiffness, limited the stress transfer mechanisms at the FRP–substrate interface.The effective bond lengths were evaluated by an experimental method, proposed by the authors, and the results were compared with what was obtained from the equation reported in the Italian Guidelines [32]. The analytical data were around 35% higher than the experimental ones; therefore, the proposed method represents a valid candidate for the experimental precautionary analysis.

Future research should be carried out to expand the experimental database on masonry units, in particular with regard to stone masonry units, in order to provide more reliable indications able to set design formulations.

## Figures and Tables

**Figure 1 materials-14-07439-f001:**
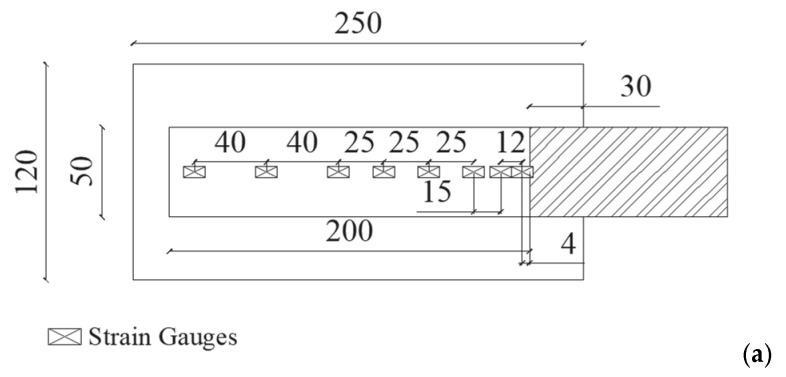
Dimensions of specimens and strain gauge setup: (**a**) setup of clay unit substrate; (**b**) set-up of clay masonry unit substrate.

**Figure 2 materials-14-07439-f002:**
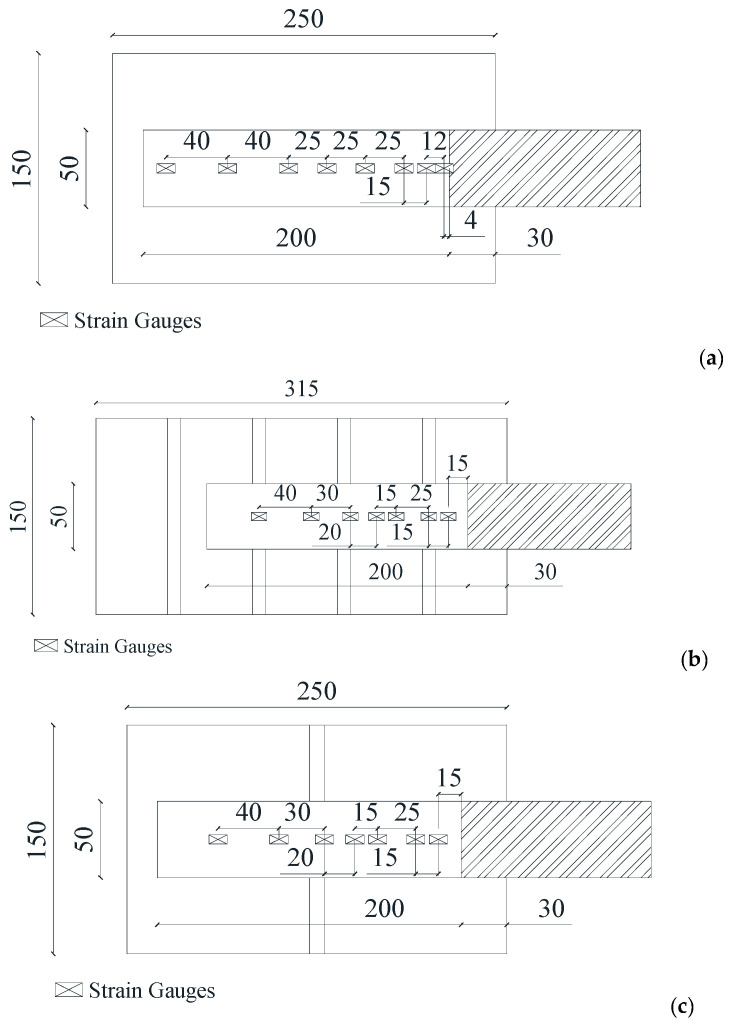
Dimensions of specimens and strain gauge setup: (**a**) stone unit; (**b**) stone masonry unit with five stone units; (**c**) stone masonry unit with two stone units.

**Figure 3 materials-14-07439-f003:**
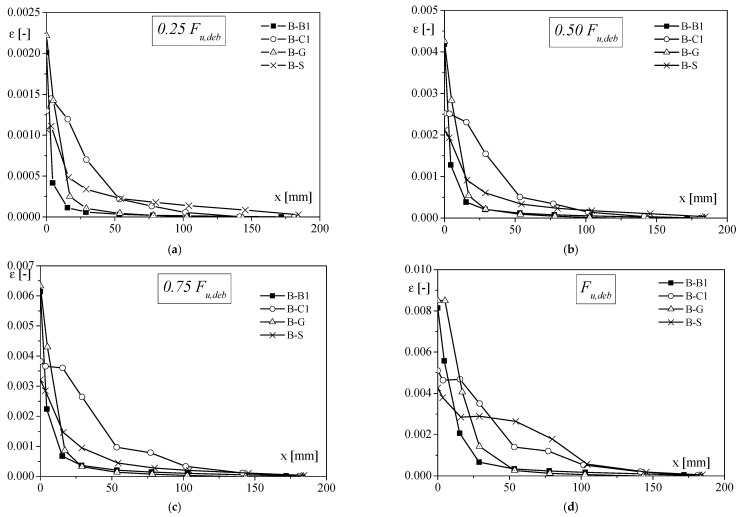
Comparison of strain distribution for specimens with clay unit substrate, varying the applied load: (**a**) F = 0.25F_u,deb_; (**b**) F = 0.50F_u,deb_; (**c**) F = 0.75F_u,deb_ and (**d**) F = F_u,deb_.

**Figure 4 materials-14-07439-f004:**
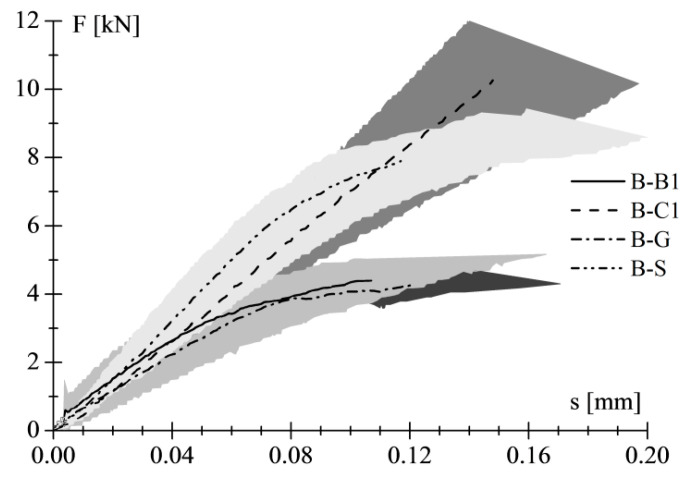
Average load–slip curves of shear bond tests on clay unit.

**Figure 5 materials-14-07439-f005:**
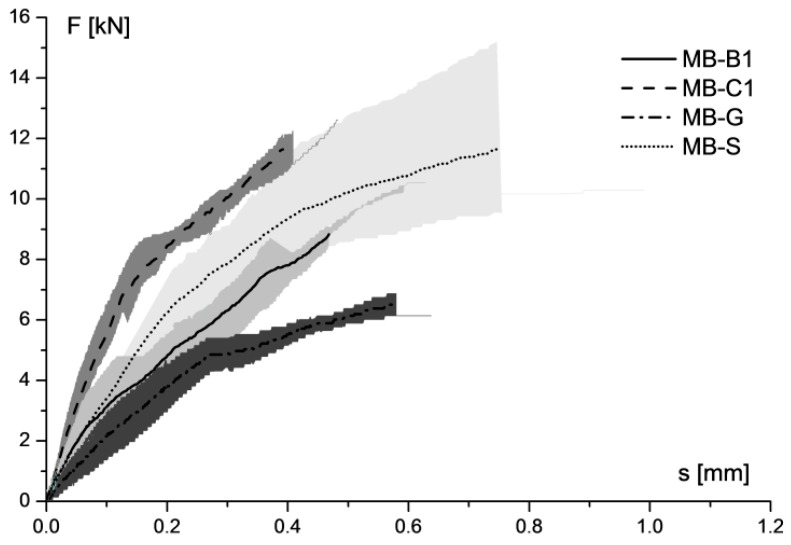
Average load–slip curves of shear bond tests on clay masonry unit.

**Figure 6 materials-14-07439-f006:**
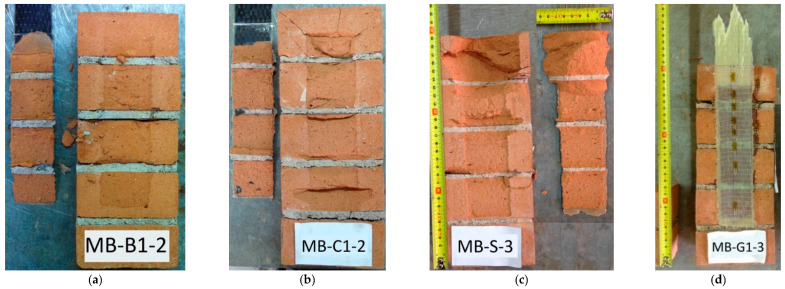
Failure mode for clay masonry unit: (**a**) MB-B1-2; (**b**) MB-C1-2; (**c**) MB-S-3; (**d**) MB-G1-3.

**Figure 7 materials-14-07439-f007:**
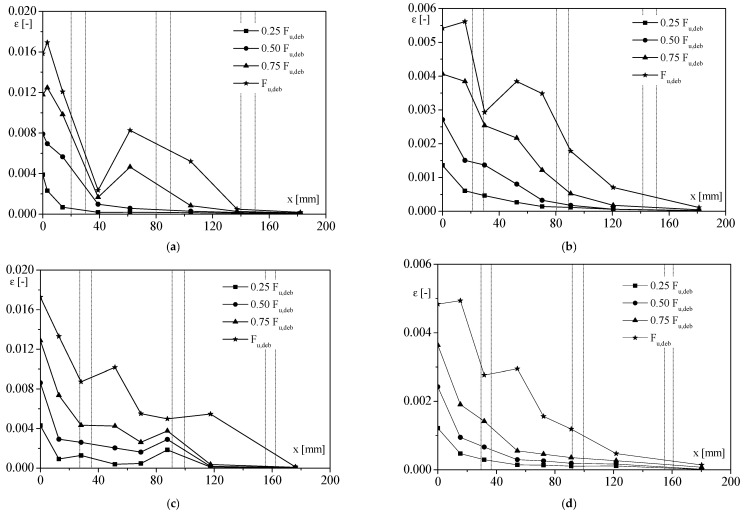
Strain distribution for samples with clay masonry unit substrate: (**a**) MB-B1-2; **(b**) MB-C1-2; (**c**) MB-G1-3; (**d**) MB-S-3.

**Figure 8 materials-14-07439-f008:**
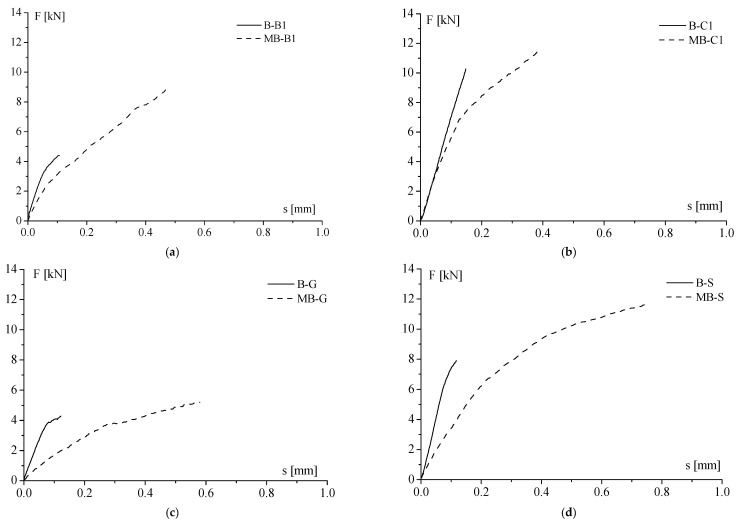
Comparison between average load–slip curves of clay unit and clay masonry, reinforced by (**a**) BFRP-1; (**b**) CFRP-1; (**c**) GFRP; (**d**) SRP.

**Figure 9 materials-14-07439-f009:**
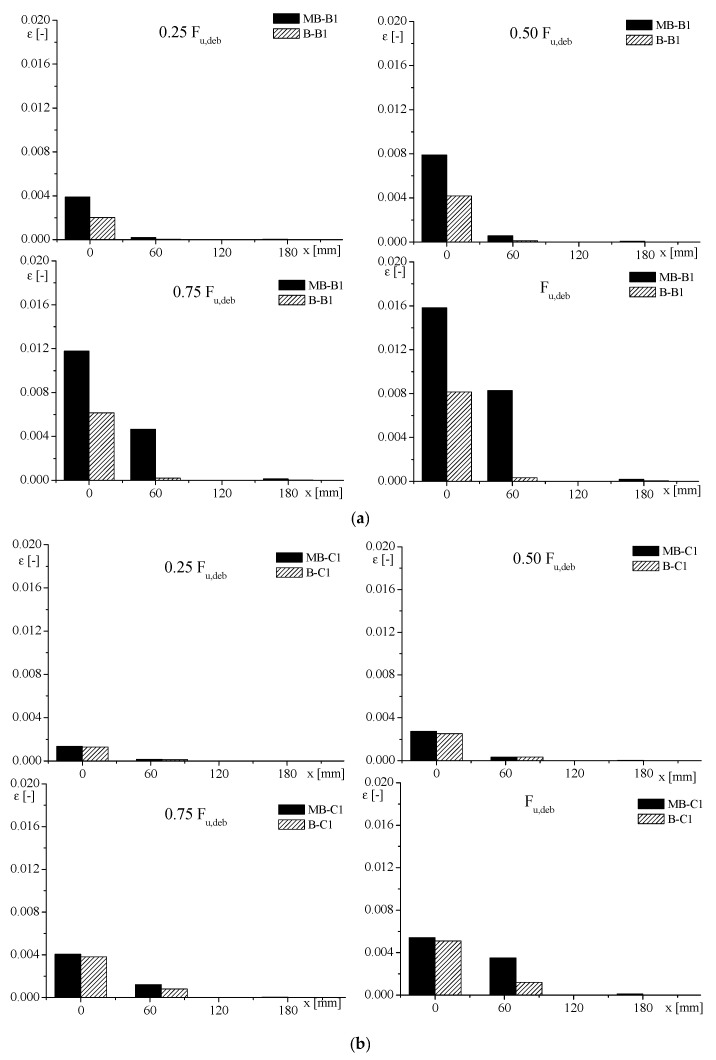
Strain–position values of clay unit and clay masonry unit with (**a**) BFRP-1; (**b**) CFRP-1; (**c**) GFRP; (**d**) SPR.

**Figure 10 materials-14-07439-f010:**
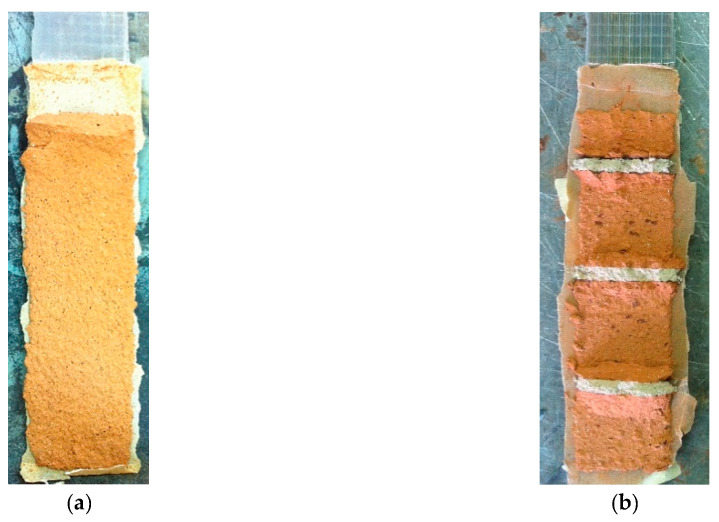
SRP detached from (**a**) clay unit; (**b**) clay masonry unit.

**Figure 11 materials-14-07439-f011:**
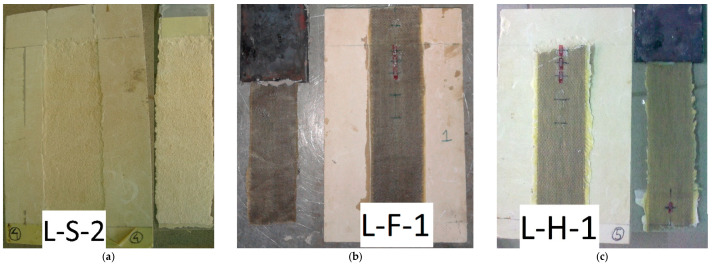
Failure mode for specimens with stone unit substrate: (**a**) L-S; (**b**) L-F; (**c**) L-H.

**Figure 12 materials-14-07439-f012:**
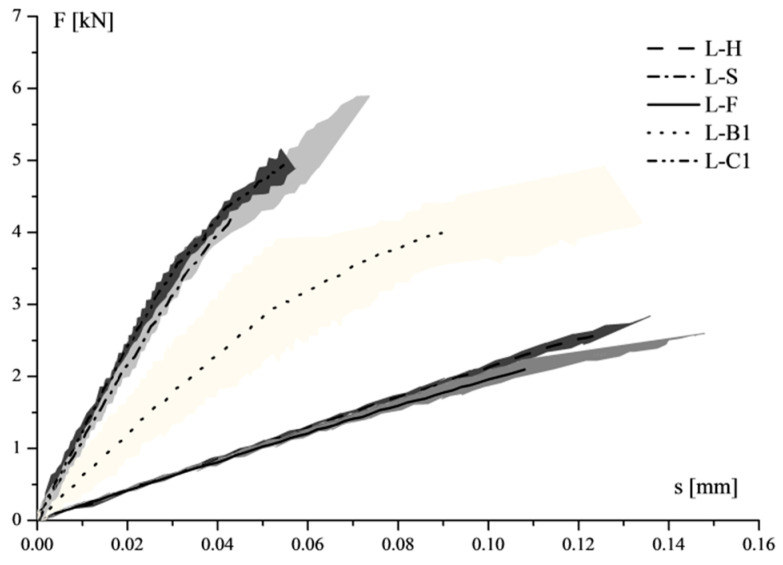
Average load–slip curves of shear bond tests on stone unit.

**Figure 13 materials-14-07439-f013:**
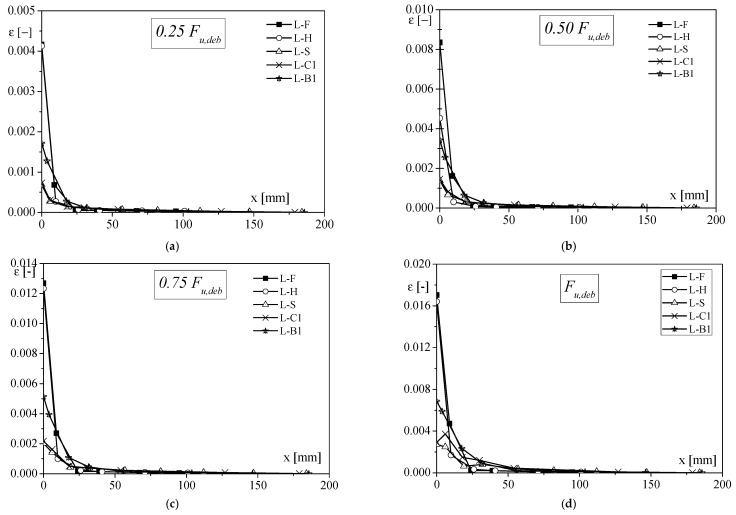
Comparison of strain distribution for specimens with stone unit substrate, varying the applied load: (**a**) F = 0.25F_u,deb_; (**b**) F = 0.50F_u,deb_; (**c**) F = 0.75F_u,deb_; (**d**) F = F_u,deb_.

**Figure 14 materials-14-07439-f014:**
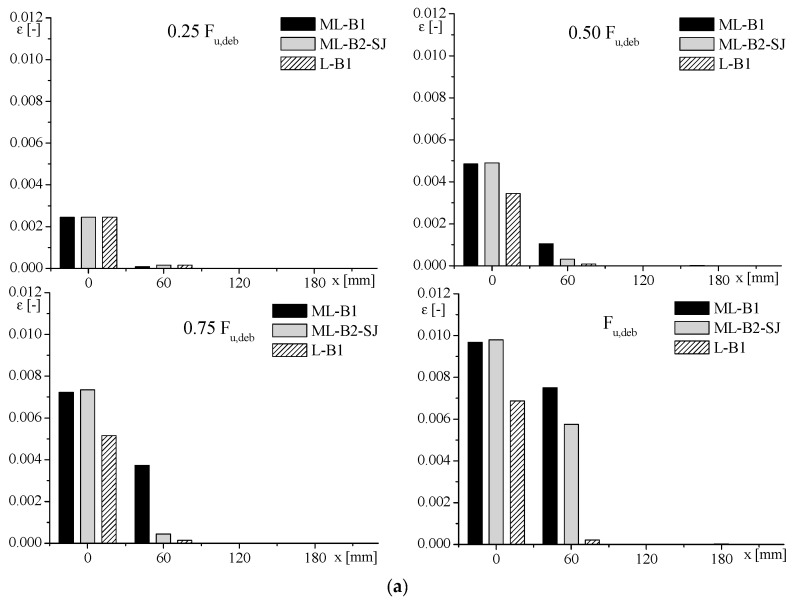
Strain–position values of stone unit and stone masonry unit substrate with (**a**) BFRP-1 and -2; (**b**) CFRP-1 and -2.

**Table 1 materials-14-07439-t001:** Mechanical properties of substrate elements.

		f_cm_ [MPa]	f_t,fm_ [MPa]	E_m_ [GPa]
Substrate 1	Clay brick unit [8]	17.89 * (5%)	3.66 (4%)	5.76 (5%)
		22.46 ** (7%)		
	Mortar1 [8]	2.62 (11%)	0.78 (25%)	5.49 (7%)
Substrate 2	Lecce Stone unit [25]	21.14 (10%)	4.26 (10%)	10.85 (16%)
	Mortar2 [25]	0.47 (12%)	0.19 (-)	-

* Compressive strength determined in perpendicular direction to the bed face. ** Compressive strength determined in parallel direction to the bed face.

**Table 2 materials-14-07439-t002:** Properties of FRP sheets.

FRP Type	f_f,t_ [MPa]	E_f_ [GPa]	ε_f,t_ [%]	t_f_ [mm]
CFRP-1	3500	240	1.45	0.170
CFRP-2	4830	230	2.00	0.166
GFRP	1400	70	2.00	0.120
BFRP-1	2000	95	2.30	0.140
BFRP-2	2000	85	2.00	0.143
SRP	3070	190	1.60	0.227
HFRP	496	22	3.10	0.155
FFRP	532	43	2.34	0.267

**Table 3 materials-14-07439-t003:** Summary of specimens tested.

Specimen	Number of Tests	Type of Substrate	Type of FRP
B-B1 °	3	Clay unit	Basalt 1
B-C1 °	2		Carbon 1
B g °	3		Glass
B-S °	3		Steel
L-B1 *	5	Stone unit	Basalt 1
L-C1 *	3		Carbon 1
L-S	3		Steel
L-H	2		Hemp
L-F	3		Flax
MB-B1	3	Clay masonry unit (4 mortar joints)	Basalt 1
MB-C1	3		Carbon 1
MB-G	3		Glass
MB-S	3		Steel
ML-B1 *	4	Stone masonry prism unit (4 mortar joints)	Basalt 1
ML-B2-SJ *	2	Stone masonry prism unit (1 mortar joints)	Basalt 2
ML-C2-SJ *	2		Carbon 2

° Already discussed in [11]. * Already discussed in [27].

**Table 4 materials-14-07439-t004:** Experimental results: clay unit substrate—BFRP, CFRP1, GFRP, and BFRP.

Sample	E_f_ t_f_ [N/mm]	F_u,deb_ [kN]	σ_u,deb_ [MPa]	η [-]	Failure Mode
B-B1	13,300	4.91 (7%)	702	0.35	Cohesive debonding and detachment of clay unit wedge
B-C1	40,800	11.46 (-)	1348	0.39	Cohesive debonding and detachment of clay unit wedge
B-G	8400	4.62 (6%)	770	0.55	Cohesive debonding and detachment of clay unit wedge
B-S	43,130	8.99 (6%)	792	0.25	Cohesive debonding and detachment of clay unit wedge

**Table 5 materials-14-07439-t005:** Experimental results: clay masonry unit substrate—BFRP, CFRP1, GFRP, and BFRP.

Sample	E_f_ ·t_f_ [N/mm]	L_b_ [mm]	F_u,deb_ [kN]	F_u,deb,ave_ [kN]	σ_u,deb_ [MPa]	η [-]	s_u,deb_ [mm]	s_u,deb,ave_ [mm]
MB-B1-1	13,300	200	10.54	9.76 (12%)	1506	0.75	0.63	0.56 (15%)
MB-B1-2			8.46		1209	0.60	0.47	
MB-B1-3			10.29		1471	0.74	0.59	
MB-C1-1	40,800	200	12.13	12.33 (2%)	1733	0.50	0.39	0.43 (11%)
MB-C1-2			12.25		1750	0.50	0.41	
MB-C1-3			12.62		1803	0.52	0.48	
MB-G-1	8400	200	7.73	6.91 (12%)	1289	0.92		
MB-G-2			6.14		1023	0.73	0.56	0.56 (-)
MB-G-3			6.87		1145	0.82	0.56	
MB-S-1	43,130	200	15.19	11.68 (26%)	1338	0.44	0.75	0.87 (20%)
MB-S-2			9.56		842	0.27	0.74	
MB-S-3			10.29		907	0.30	0.99	

**Table 6 materials-14-07439-t006:** Experimental results: stone unit and stone masonry unit—BFRP, CFRP1, and CFRP2 reinforcement [28].

Sample	E_f_ t_f_ [N/mm]	F_u,deb_ [kN]	σ_u,deb_ [MPa]	η [-]	Failure Mode
L-B1	13,300	4.35 (9%)	622	0.31	Cohesive debonding and shear cracks in the substrate
L-C1	40,800	4.58 (17%)	539	0.15	Cohesive debonding
ML-B1	13,300	6.57 (21%)	939	0.47	Cohesive debonding, detachment of stone unit wedge and shear cracks in the substrate
ML-B2-SJ	12,155	5.14 (-)	718	0.15	Cohesive debonding and detachment of stone unit wedge
ML-C2-SJ	38,180	6.85 (-)	825	0.17	Cohesive debonding and shear cracks in the substrate

**Table 7 materials-14-07439-t007:** Experimental results: stone unit—SFRP, HFRP, and FFRP.

Sample	E_f_ ·t_f_ [N/mm]	L_b_ [mm]	F_u,deb_ [kN]	F_u,deb,ave_ [kN]	σ_u,deb_ [MPa]	η [-]	s_u,dev_ [mm]	s_u,deb,ave_ [mm]
L-S-1	43,130	200	6.01	5.08 (17%)	530	0.17	0.07	0.06 (27%)
L-S-2			4.91		433	0.14	0.06	
L-S-3			4.30		379	0.12	0.04	
L-F-1	11,481	200	2.44	2.42 (8%)	183	0.34	0.14	0.13 (16%)
L-F-2			2.59		194	0.36	0.15	
L-F-3			2.22		166	0.31	0.11	
L-H-1	3410	200	2.47	2.65 (-)	318	0.64	0.12	0.13 (-)
L-H-2			2.83		366	0.74	0.14	

**Table 8 materials-14-07439-t008:** The values of effective bond length.

Sample	l_ek,exp_	l_ek,1_	l_ek_	l^*^_ek,1_	l^*^_ek_
	[mm]	[mm]	[mm]	[mm]	[mm]
B-B1	98 (16%)	99	150	134	150
B-C1	147 (-)	174	174	234	234
B-G	109 (20%)	80	150	109	150
B-S	173 (1%)	186	186	251	251
L-B1	74 (12%)	114	150	119	150
L-C1	139 (7%)	200	200	209	209
L-S	140 (6%)	205	205	215	215

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
