# Peer review of "Bond Tests on Clay Bricks and Natural Stone Masonry Externally Bonded with FRP"

_materials, 2021, doi:10.3390/ma14237439_

Round 1

Reviewer 1 Report

  1. The conclusion of the abstract is not well summarized, so it is difficult to quickly read the main conclusions of this paper.
  2. Although most of the test methods are given in references 8, 9 and 25, I still suggest to make a basic explanation in this paper, such as the setting of sensor spacing, the setting of anchorage length, etc.
  3. Figures 8 and 9 are the core of this article. They illustrate that with increasing the stiffness of FRP a lower influence of the presence of mortar joints is found in terms of strains. I suggest adding quantitative analysis here.
  4. The effects of  the substrate,  the mortar joints  and the rigidity of FRP are discussed in this paper ,  but lacks a systematic summary. I suggest  the authors add a summary in the discussion section

Author Response

The authors would like to thank the reviewer for the positive comments on their manuscript. All the suggestions provided by the reviewer in the next points are included in the revised manuscript and significantly contributed to improve the quality of the manuscript

Reviewer 2 Report

The research article "Bond tests on clay bricks and natural stone masonry externally bonded with FRP" is interesting. There is need to revise few things;

  1. Abstract need more information regarding metholodgy and research parameters.
  2. Different strengthening methods have been used in the past for repair of masorny construction. Authors are suggested to included in the introduction section as well.
  3. Also, different hollow blocks and cement clay interlocking brick walls have been studied. There is need to review and consider them in introduction.
  4. Section 2.1 is merged with many informations, its suggested to subdived this section to properly indicate test matrix, materials properties, size of test specimens, and loading setup. 
  5. Similary once again there is section 2.2. In general both sections need to re-arrange properly. 
  6. Further many results are presented in a single section 3.1, which is not appropriate. Failure modes, load verus slip curves and other parameters could be properly discussed in separate sections.
  7. Conlcusions are not clear. Conclusion section should be concise and to the point.

Author Response

(The authors gave the same response as above.)

Reviewer 3 Report

The paper is written correctly and presents a current engineering issue. However, for it to be published, in my opinion, a few things need to be improved:

  1. The introduction should be strengthened with references concerning testing of CFRP-type materials: 10.1007/978-3-030-84958-0_33, 10.1016/j.tws.2020.106627, 10.1007/978-3-030-80312-4_21.
  2. Please clearly demonstrate in the introduction the novelty of this paper in relation to other thematically similar research papers.
  3. Figure 8 should be larger - it has too much information at the current size. Moreover, this Figure should be present in different way, in order to better present geometrical parameters. The same situation applies to the Figure 2.
  4. Please refer in the conclusion to the quantitative assessment of the research conducted - as this information is missing.

Author Response

(The authors gave the same response as above.)
